# Corneal Endothelial-like Cells Derived from Induced Pluripotent Stem Cells for Cell Therapy

**DOI:** 10.3390/ijms241512433

**Published:** 2023-08-04

**Authors:** Xiao Yu Ng, Gary S. L. Peh, Gary Hin-Fai Yam, Hwee Goon Tay, Jodhbir S. Mehta

**Affiliations:** 1Tissue Engineering and Cell Therapy Group, Singapore Eye Research Institute, Singapore 169856, Singapore; ng.xiao.yu@seri.com.sg (X.Y.N.); gary.peh.s.l@seri.com.sg (G.S.L.P.); gary.yam@pitt.edu (G.H.-F.Y.); 2Ophthalmology and Visual Sciences Academic Clinical Program, SingHealth and Duke-NUS Medical School, Singapore 169857, Singapore; hweegoon.tay@duke-nus.edu.sg; 3Corneal Regeneration Laboratory, Department of Ophthalmology, University of Pittsburgh, 6614, Pittsburgh, PA 15260, USA; 4Centre for Vision Research, DUKE-NUS Medical School, Singapore 169857, Singapore; 5Department of Cornea and External Eye Disease, Singapore National Eye Centre, Singapore 168751, Singapore

**Keywords:** iPSC, corneal endothelial, cell therapy

## Abstract

Corneal endothelial dysfunction is one of the leading causes of corneal blindness, and the current conventional treatment option is corneal transplantation using a cadaveric donor cornea. However, there is a global shortage of suitable donor graft material, necessitating the exploration of novel therapeutic approaches. A stem cell-based regenerative medicine approach using induced pluripotent stem cells (iPSCs) offers a promising solution, as they possess self-renewal capabilities, can be derived from adult somatic cells, and can be differentiated into all cell types including corneal endothelial cells (CECs). This review discusses the progress and challenges in developing protocols to induce iPSCs into CECs, focusing on the different media formulations used to differentiate iPSCs to neural crest cells (NCCs) and subsequently to CECs, as well as the characterization methods and markers that define iPSC-derived CECs. The hurdles and solutions for the clinical application of iPSC-derived cell therapy are also addressed, including the establishment of protocols that adhere to good manufacturing practice (GMP) guidelines. The potential risks of genetic mutations in iPSC-derived CECs associated with long-term in vitro culture and the danger of potential tumorigenicity following transplantation are evaluated. In all, this review provides insights into the advancement and obstacles of using iPSC in the treatment of corneal endothelial dysfunction.

## 1. Background

The human corneal endothelium (CE) is a monolayer of hexagonally shaped cells located in the innermost layer of the cornea. The CE regulates corneal hydration and corneal transparency through a dynamic interaction at a leaky barrier of tight junctions and active ionic pumps [1,2,3,4,5,6]. Human corneal endothelial cells (CECs) are mitotically inactive and have a limited capacity for cellular regeneration, causing a significant loss of CECs due to detrimental diseases or trauma. To maintain the functional integrity of the CE, remaining CECs spread and enlarge to compensate for the missing cells [7,8]. However, if the endothelial cell density (ECD) falls below a critical threshold (Figure 1), the dynamic balance of corneal hydration will be inefficient, leading to the decompensation of the CE and the development of stromal edema [3,8]. As a consequence, the cornea becomes cloudy, visual acuity is diminished, and eventually lead to corneal blindness if left untreated (Figure 1) [1].

Corneal endothelial dysfunction contributes to a significant proportion of corneal blindness (Figure 2A) [9,10,11], out of which, Fuchs endothelial corneal dystrophy (FECD) and pseudophakic bullous keratopathy (BK) are the leading indications [9,10]. A Global Survey of Corneal Transplantation and Eye Banking conducted from 2012 to 2013 estimated that approximately 12.7 million people were waiting for transplantation due to a shortage of donor graft material. Even in countries with adequate donor supply, the median waiting time for corneal transplant (CT) was 6.5 months. The United States had the highest transplantation rate, with 199.10^−6^ CT per capita, while the median among the 116 countries investigated was only 19.10^−6^ [7]. The survey also revealed that 53% of the world’s population had no access to corneal transplantation [7], highlighting a significant disparity in treatment accessibility for patients awaiting corneal transplantation.

The current practice for treating CE dysfunction is primarily dominated by endothelial keratoplasty (EK) (Figure 2A,B) or, to a lesser extent, penetrating keratoplasty (PK), both of which depend entirely on the availability of suitable donor corneal tissues [13]. With an ever-increasing demand for corneal transplantation exacerbated in part by a rapidly aging population enjoying longer life spans [14], compounded by a shortage of donor graft material, there is a pressing need to explore alternative forms of treatment such as cell-based therapeutics. Such therapies offer the potential for greater sustainability and the possibility of alleviating the demand for corneal transplants [15]. Significant progress has been made towards utilizing cultured donor corneal endothelial cells (CECs) for transplantation [16,17,18]. However, it is important to note that this method does not completely eliminate the need for donors, but rather reduces the requirement. Additionally, primary CECs have a limited capacity for in vitro expansion [17]. Hence, embryonic stem cells (ESCs) and induced pluripotent stem cells (iPSCs) are alternative sources as they possess self-renewing capabilities, allowing for indefinite propagation. More importantly, these pluripotent cells have the capacity to differentiate into all cell types of the three embryonic germ layers including corneal endothelial cells [19,20]. Between the two, the use of iPSCs circumvents ethical debates surrounding the use of pluripotent ESCs, as they do not require the destruction of embryos or oocytes [21]. Moreover, iPSCs offer a more versatile solution, as they can be derived from various sources of adult somatic cell types.

## 2. Available Methods of Treatment for Corneal Endothelial Dysfunction

Penetrating keratoplasty (PK) is a surgical procedure whereby the entire cornea is replaced. It has been the primary technique for CT for the past five to six decades [9,22]. However, in recent years, there has been rapid development and increased popularity of lamellar grafts, particularly posterior lamellar grafts, also known as endothelial keratoplasty (EK), which has gained significant traction, accounting for approximately 60% of grafts registered in Australia and United States [9,10,23]. Singapore has also shown a substantial adoption rate for EK, accounting for 44% of keratoplasty procedure and rising [24].

In contrast to full-thickness PK surgery, EK is the selective replacement of the dysfunctional CE [11]. Specifically, Descemet’s stripping endothelial keratoplasty (DSEK), Descemet’s stripping automated endothelial keratoplasty (DSAEK) (Figure 3B), and Descemet’s membrane endothelial keratoplasty (DMEK) (Figure 3A) can achieve higher surgical success rates and has become the preferred choice for corneal endothelial replacement [1,22]. According to the Australian Corneal Graft Registry report in 2020, DMEK comprised 31% of registered techniques, while DSEK/DSAEK accounted for 29% of all registered techniques [10]. Similarly, in the United States, both DMEK and DSEK/DSAEK procedures represented approximately 30% each of the total keratoplasty procedures performed in 2022 [23].

Current CT surgeries are faced with multiple challenges, including a global shortage of suitable donor graft materials, hindering the ability to meet patient demands. The long-term survival of transplanted donor grafts is associated with the age of the donor, and is significantly influenced by ECD (Figure 4) [10]. Grafts from donors with fewer than 2500 cells/mm^2^ (generally older donors 50 years and above) have significantly poorer survival rates compared to those with higher corneal ECD (from younger donors below 30 years of age) [10]. Additionally, issues like primary graft failure not only exacerbate this demand for CT, it has also been shown that repeated grafts in such cases exhibit significantly lower survival rates than initial grafts [10]. Collectively, the above issues highlight the limitations of relying solely on donor graft materials for corneal transplantation. Consequently, researchers are exploring alternative treatments in the field of cellular therapeutics. One such approach involves delivering functional primary human corneal endothelial cells (CECs) isolated from donor corneas that have been expanded in culture.

The techniques for in vitro culture of human CECs have significantly improved over the last 15 years, enabling the limited expansion of bona fide functional primary cultures of CECs [16,17,18]. Culture conditions have been optimized, such as the dual media culture system, which involves the use of a proliferative F99-M4 media for the propagation of CECs, followed by subsequent stabilization in Endo-M5 media to preserve their cellular morphology [17]. The expanded cells were shown to be functional within a rabbit model of BK [18]. Using a different culture approach based on a series of Opti(MEM) media formulations [16], Kinoshita et al. injected the expanded primary CECs in patients recruited in a human clinical trial (International Clinical Trials Registry Platform: JPRN-jRCTa050190118) [26], and they showed promising outcomes [27] (more below). In their study, primary CECs were expanded in a series of media supplemented with different small molecules including Rho-associated protein kinases (ROCK) inhibitor Y-27632, p38 MAP kinase inhibitor SB-203580, and transforming growth factor beta inhibitor SB-431542 [16].

The characterization of the propagated primary CECs is an important aspect of quality assurance and is necessary to ensure their suitability for therapeutic applications [13]. This is generally accomplished with techniques such as real-time PCR and immunocytochemistry, focusing on specific markers associated with corneal endothelial function [28]. Commonly assessed genes include collagen type VIII, alpha 1 (*COL8A1*), and Solute Carrier Family 4 Member 11 (*SLC4A11*) [17]. The immunocytochemistry characterization of CECs often involves examining markers including sodium–potassium-transporting adenosine triphosphatase (*Na^+^K^+^-ATPase*) and zona occludens 1 (*ZO-1*) [17]. Flow cytometry enables the use of cell-surface markers for the characterization of CECs. For instance, the negative or low expression of *CD44*, *CD24*, *CD26*, *CD105*, and *CD133* and the positive expression of *CD166* and TAG2A12 (*sPRDX-6*) have been identified as valuable markers [16,22,29,30]. More will be elaborated below.

Currently, two main approaches have been described for the delivery of expanded primary CECs; the first is via cell injection, and the second as tissue-engineered graft material. In the cell-injection approach, the expanded CECs are directly injected into the anterior chamber of the patient’s eye [9,16,18], and this has been successfully carried out in human trials [16,27]. In the aforementioned study, a group of eleven patients with bullous keratopathy underwent an injection of approximately 1 × 10^6^ expanded CECs into the anterior chamber, and the patients were subsequently placed in a prone face-down position for three hours [16]. Remarkably, all eleven patients maintained corneal transparency during the two-year follow-up period [16]. A more recent publication of the five-year follow-up on the patients revealed that ten out of the eleven eyes exhibited normal corneal endothelial function with no significant adverse reactions observed [27].

Bandeira et al. recently described an innovative study involving the injection of cultured CECs embedded with magnetic nanoparticles into a rabbit corneal endothelial dystrophy model [31]. The presence of magnetic nanoparticles enabled a more precise delivery of injected cells using magnetic field guidance, resulting in a greatly reduced time needed for the rabbit to remain in a prone position compared to conventional cell injection (30 min vs. 3 h) [31]. Notably, injected CECs without magnetic nanoparticles exhibited a more heterogeneous cellular distribution, whereas those with the nanoparticles settled centrally within the wound area [31]. This centralized distribution is advantageous as it has the potential to accelerate the recovery process and enhance the overall efficacy of the procedure [31]. A separate group is currently conducting an ongoing clinical trial (ClinicalTrials.gov: NCT04191629) sponsored by Emmecell to evaluate the efficacy of a magnetic cell delivery nanoparticle platform for delivering CECs to patients with corneal edema [32]. In this trial, CECs loaded with magnetic nanoparticles are injected into the patient’s eye and guided to the appropriate site using an external magnetic patch placed over the eye [32]. These innovative magnetic variants to cell injection offer the potential for improved targeted delivery of CECs and may eventually enhance the treatment outcomes for patients receiving corneal endothelial cell injection therapy [31,32].

The second approach for delivering cultured CECs has been shown to be possible using tissue-engineered grafts. For this, the propagated CECs are seeded onto a scaffold carrier, such as a thin decellularized stromal lenticule with an intact Descemet’s membrane (DM). This technique has demonstrated its effectiveness in a proof-of-function study in a rabbit BK model, where it successfully reversed corneal blindness in the rabbits that receive the tissue-engineered grafts [28,33]. Additionally, other biomedical material such as collagen sheet [34], agarose-based membrane [35], and bio-adhesive gelatin disc [36] have also been described as potential delivery constructs suitable for tissue engineering, highlighting the potential of tissue engineering in facilitating corneal endothelial cell transplantation.

Regardless of the delivery approach, corneal endothelial cell-based therapeutics using primary CECs still require cadaveric donor corneas for its isolation and expansion. Moreover, there is a limit to the capacity of primary cell expansion due largely to cellular senescence [37,38]. This issue is further exacerbated with worldwide trends of increasing life expectancy and aging populations [14], which may further widen the supply–demand mismatch, highlighting the need for an alternative cell source not reliant on donor corneas.

Pluripotent stem cells, whether embryonic or induced, are promising sources of cells for cultivating CECs due to their capacity for self-renewal. Given the correct differentiation cues, they have the ability to form derivatives of the three embryonic germ layers—the ectoderm, mesoderm, and endoderm [19,20]. Numerous methods have been described for the differentiation of CECs from both ESCs and iPSCs, and these stem cell-derived CECs would be the ideal cell source for cell-based therapeutics for treating corneal endothelial dysfunction [39]. The derivation of pluripotent human ESCs from the inner cell mass of a human blastocyst was first described in 1998, these cells can give rise to all cell types within the human body [19,20]. Although the potential of ESCs is widely described, with multiple embryonic stem cell lines established [40], research using ESCs is shrouded by moral concerns as an embryo has to be destroyed for the derivation of embryonic stem cells [41].

With the discovery of iPSCs by Takahashi and Yamanaka in 2006 [42], it became possible to reprogram terminally differentiated adult somatic cells back to a pluripotent state by introducing transcription factors *OCT4*, *SOX2*, *KLF*, *c-MYC*, *NANOG*, and *LIN28* [43]. Similarly to ESCs, iPSCs possess the remarkable differentiative potential to give rise to all cell types of the three embryonic germ layers and exhibit similar gene expression patterns and epigenetic profiles [44].

Leveraging iPSCs in research provides a valuable platform for investigating underlying disease mechanisms and evaluating specific therapeutic interventions. The generation of disease-specific cell lines using iPSCs enables in vitro disease modelling, providing insights into disease pathology [44]. One advantage of iPSCs is the ability to generate patient-specific iPSCs, which supports the concept of ‘personalised medicine’ and reduces the reliance on donor corneas [45]. The reprogramming technology used in the generation of iPSCs continues to advance, facilitating the production of high-quality iPSC lines [46]. For example, some of these advancements made towards more efficient derivation of iPSCs include improvements made to the media formulation through the use of substrates and small molecules [45,47,48,49]. The reprogramming process has evolved from a reliance on integration-dependent viral systems to integration-independent systems [45,50,51], thereby minimizing the risk of genomic instability and chromosomal aberrations resulting from the integration of exogenous genes into the host genome [45,50]. In particular, the development of integration-independent systems such as the recombinant Sendai virus (SeV) vector has resolved these concerns by enabling the expression of reprogrammed genes without chromosomal integration [45,51].

## 3. iPSCs Are Fast Becoming the Holy Grail

Following the groundbreaking discovery of iPSCs, there has been an explosion of knowledge and remarkable progress in the understanding of diseases and disorders at the cellular level, leading to advancements in the field of cell-based regenerative medicine [42,44,52,53]. Clinical trials have been initiated, and human iPSC-derived cellular therapeutic products are being evaluated for their efficacy and safety [54,55].

A search on NCBI PubMed using the term “Induced pluripotent stem cells” yields a total of 28,241 entries. The number of papers relating to “induced pluripotent stem cells” has exponentially risen, particularly following the discovery of the Yamanaka factors (*Oct3/4*, *Sox2*, *c-Myc*, and *Klf4*) in 2006 [42]. The number of papers published went from 67 in 2006 to an average of 3102 papers per year for the period 2020 to 2022 [56]. The surge in publications reflects the immense interest in exploring the full potential of iPSCs in various areas, including disease modelling and therapeutics. A refined search using the keywords “iPSCs” and “corneal endothelial” yields 42 results, with 29 publications in which the researchers are actively working on establishing workflows and refining protocols specifically relevant to corneal endothelial-based research, out of which 10 are review papers [56]. This shows that the use of iPSCs for the treatment of corneal endothelial dysfunction is still in its infancy for the generation of CECs from iPSCs.

## 4. Progression in the Technology

Most approaches to differentiating corneal endothelial cells from iPSCs involve either reprogramming adult somatic cells into iPSCs or directly differentiating them from established human iPSC lines. The first approach utilizes an integration-independent system to reprogram somatic cells into pluripotent stem cells. As technology in this field continues to advance, commercially available reprogramming kits such as the CytoTune-iPS Sendai Reprogramming kit (Thermo Fisher Scientific, Waltham, MA, USA) [57,58,59,60] utilize Sendai particles to deliver the Yamanaka factors into somatic cells, reprogramming them into iPSCs. The second approach involves the direct induction of CECs from human iPS cell lines, which can now be obtained commercially (Applied StemCell-ASC, ATCC) or through organizations supporting iPS cell research (Center for iPS Cell Research and Application-CiRA Foundation, European Bank for induced pluripotent Stem Cells-EBiSC) [61,62,63,64]. There are also clinical-grade iPS cell lines available that follow strict GMP guidelines and comply with FDA regulations [64].

Various commercially available extracellular matrices can be used for the culture of iPSCs, including Corning Matrigel^®^ matrix, and, more recently, recombinant proteins like vitronectin and laminin (Laminin-511, Laminin-521). These substrates provide an optimal surface for the cellular attachment of iPSCs and promote their survival, leading to long-term self-renewal [65,66]. Studies have demonstrated that Laminin-521 coating supports efficient expansion of iPSCs within a closed cell expansion system, resulting in the production of a greater number of iPSCs in a shorter period, thereby ensuring a stable supply of iPSCs for differentiation into other cellular therapeutic products [67].

### 4.1. Differentiation from iPSCs to NCCs

The corneal endothelium arises from the periocular mesenchyme (POM), which consists of cranial NCCs [68,69,70]. During ocular development, the initial wave of POM cells migrates between the presumptive corneal epithelium and lens, forming the endothelial cell layer [68,69,70]. Developmentally, the corneal endothelium forms from the NCCs [68,69,70], hence differentiating the iPSCs first into NCCs and, subsequently, into CECs, mimics the natural cell-fate specification of CECs [71]. Babushkina et al. reported that periocular neural crest cells (pNCs) are multipotent, and the nascent cornea can induce injected pNCs into various ocular tissues, including corneal endothelium [70]. The injected pNCs up-regulated genes that are involved in the development of the chick corneal endothelium [70]. This suggests that NCCs drive the differentiation of CECs, and that local cues play an important role in defining NCC-to-CEC fate. Hence, emulating the natural CEC fate specification via directed NCCs contributes to a robust cell source of CECs [71]. Zhao et al. demonstrated that restricting neural crest fate toward the ocular lineage produced iPSC-derived CECs that resemble CECs morphologically and expressed typical CEC markers [71].

Overall, there is a lack of knowledge of which transcription factors are at work in NCC-migration and CEC-maturation, which greatly limits the ability to directly differentiate iPSCs to CECs. Some of the transcription factors involved in the mechanism are as follows: *Foxc1*, *Foxc2*, *Lmx1b*, *Pax6*, *Pitx2*, *RARβ*, *RARγ*, *RXRα*, *Six3*, and *Smad2* [39,72,73,74,75,76,77]; however, it has not been established as to which of them are critical in NCC migration or in CEC maturing process [39]. Furthermore, available markers currently used to characterize CECs that are unique to CECs are very limited (discussed later), which complicates the ability to determine differentiation efficiency; Hatou et al. differentiated iPSCs directly into Corneal Endothelial Cell Substitute cells (CECSi), but due to the lack of CEC-specific markers, they had difficulty characterizing these cells [61]. With the availability of CEC-specific transcriptomic dataset [78], leading to better understanding of the transcriptome of CECs, there may be more research groups developing direct differentiation protocols [61], thus making the differentiation process more efficient and straightforward.

Currently, most research groups have chosen to differentiate iPSCs to CECs via the directed method through NCCs. The most commonly used method for generating NCCs from iPSCs is the use of a chemically defined medium along with small molecules. The differentiation of iPSCs towards NCCs is depicted in part in Figure 5. McCabe et al. described a “dual Smad inhibition” approach for the derivation of NCCs from ESCs by using a TGF-beta signalling blocker (SB-431542) and a BMP inhibitor (Noggin) to target both TGF-beta-Smad-2/3 and BMP-Smad-1/5/8 signalling pathways [79]. Ali et al. adopted the “dual Smad inhibition” strategy and successfully generated NCCs from iPSCs as well [59]. Menendez et al. describes a protocol whereby iPSCs were differentiated into NCCs by combining Smad inhibition with the activation of the Wnt pathway, resulting in a highly efficient single-step generation of neural crest stem cells from iPSCs [80]. This was performed in a chemically defined medium using commercially available inhibitors of TGF-β signalling and glycogen synthase kinase 3 (GSK-3) activity. Jia et al. adapted this protocol and was able to differentiate iPSCs into NCCs by modulating TGF-β and Wnt signalling pathways by incorporating a TGF-β inhibitor (SB-431542) as well as a GSK-3 inhibitor/WNT activator (CHIR99021) into the differentiation process [62].

### 4.2. Differentiation from NCCs to CECs

Although the differentiation of iPSCs to NCCs has become more straightforward with refined protocols, the process of directing the differentiation of NCCs toward CECs is more complex due to the mechanisms involved in neural crest cell migration and corneal endothelial cell maturation (Figure 5) [39]. The migration of NCCs begins with epithelial–mesenchymal transition (EMT), and this process is believed to involve TGF-β signalling and Wnt signalling [39,73,81]. However, CECs ultimately acquire an endothelial morphology, and hence, a mesenchymal–endothelial transition (MET) is likely to play a role in the maturation of the corneal endothelium [39]. Bosch et al. differentiated NCCs into CECs using a conditioned medium from human CECs [57]. While this method provided the advantage of exposing the NCCs to CEC-specific signalling factors within the conditioned medium, the undefined composition of the medium makes it difficult to identify the specific component(s) responsible for inducing differentiation [57,62].

A fully defined xeno-free medium for CEC differentiation will be more suitable for translation towards clinical applications [62]. Specifically, a feeder-free, serum-free induction method using recombinant protein or small-molecules is a more feasible approach. Jia et al. reported the differentiation of NCCs towards CECs using a combination of B-27 supplement, PDGF-BB, and a WNT-inhibitor XAV-939 [62]. In that study, the use of serum was replaced with a chemically defined lipid concentrate [62]. Similar approaches include the use of knock-out serum replacement in addition to inhibitors or activators of specific pathways such as DKK-2 (Wnt inhibitor), SB-431542, CHIR-99021 [58,59,60,63,82], or a combination of small molecules such as A-769662 (AMPK activator) and AT-13148 (Akt inhibitor) [82]. Hatou et al. described an induction approach to directly differentiate iPSCs by regulating the gene expression levels of *C-JUN*, *C-FOS*, *NR3C2*, and *SOX9* using a combination of optimized cytokines in a chemically defined, animal and human origin-free corneal endothelial differentiation medium (CEDM) [61]. A summary of the different iPSC-to-CEC induction protocols are available in Appendix A.

A plausible approach to direct the differentiation of iPSCs can be via 3D spheroid cultures, and this can be adaptable for the differentiation of CECs from iPSCs. Shah et al. reasoned that 3D spheroid culture enables cell–cell and cell–ECM interactions, thereby allowing the cells to grow in a biochemical environment that closely resembles the in vivo environment [83]. This culture system is also more receptive to cellular enrichment, providing a larger population of useable and desired cell type with appropriate selection [84]. The 3D spheroidal differentiation approach below originated from the differentiation of photoreceptor precursor cells, where iPSCs were cultured on ultra-low adhesion sphere-forming plates and differentiated in media supplemented with Y-27632 and IWR1e (WNT inhibitor) [84]. Sequentially, the differentiation process involved adding a 1% ECM mixture of human type 1 and type 3 collagen, vitronectin, and fibronectin, and subsequently re-seeding the expanding iPSC spheres onto a 100 mm ultra-low attachment culture dish in the next phase of the differentiation process [84]. This medium was supplemented with 1% ECM, 3 mM CHIR-99021, and 100 nM smoothened agonist (SAG), before switching to a neural retina culture medium [84]. Based on this study, as well as studies demonstrating formation of functional 3D spheres of CE precursor cells derived from rabbit CE [85] and from cultured human CECs [86] within a rabbit bullous keratopathy model [85,86], exploring the concept of generating iPSC-derived CECs using 3D spheroid differentiation is indeed a plausible avenue to pursue.

**Figure 5 ijms-24-12433-f005:**
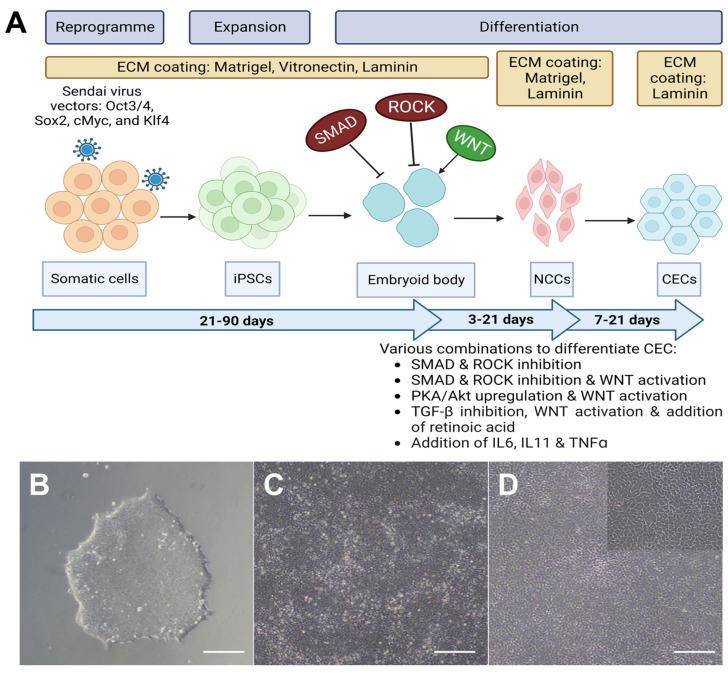
(**A**) Schematic diagram of the differentiation process for deriving human CEC from iPSC, created with BioRender.com. (**B**) Brightfield image of an iPSC colony [82]. (**C**) Brightfield image of NCSCs/NCCs [82]. (**D**) Brightfield image of iPSC-derived CECs [82]. Scale bar (**B**–**D**): 200 μm, (**D**) magnified image: 50 μm.

### 4.3. Characterization of iPSC-Derived CECs

As described above, different approaches are utilized to drive the differentiation of iPSCs towards a CEC identity. Hence, the characterization of these iPSC-derived CECs becomes extremely crucial in determining the success of the differentiation protocols. The characterization of iPSC-derived CECs must include a combination of assays including but not limited to morphological assessment, real-time PCR, immunohistochemistry, flow cytometry, as well as karyotype analysis. The process of characterization should begin from the induction of iPSCs to NCCs, and all the way till the CEC differentiation is deemed complete. Some of the markers that are often tested at the pluripotency stage include *OCT3/4*, *SSEA-3*, *SSEA-4*, *SOX2*, *GCTM-2*, *TRA-1-60*, and *Nanog*; those tested at the neural crest stem cell (NCSCs) stage include *AP2α (TFAP2A)* and *Nestin*, NCC markers (*SOX9*, *SOX10*, *NGFR*, *HNK-1*, *Vimentin*) and human CEC markers (*Na^+^K^+^-ATPase*, *ZO-1*, *SLC4A11*, *AQP1*, *N-cadherin*, *COL4A1*, *COL8A1*, *COL8A2*) [17,57,58,59,60,62,63,82].

The most widely described markers associated with CE functions include *Na^+^K^+^-ATPase*, *ZO-1*, *SLC4A11*, and Type VIII collagen (both *COLO8A1* and *COL8A2*). *Na^+^K^+^-ATPase* is an essential enzyme that actively transports sodium ions out of the CECs and potassium ions into the cells, establishing an osmotic gradient that facilitates fluid movement out of the cornea [87]. The osmotic gradient moves excess fluid out of the cornea, thereby preventing edema or swelling. *ZO-1* is a tight junction protein involved in signal transduction at cell–cell junctions and forms a “leaky” barrier that contributes to the regulation of corneal hydration, and its co-expression with *Na^+^K^+^-ATPase* indicates the dynamic regulation and the intricate balance behind corneal transparency [88]. *SLC4A11*, a transmembrane protein, plays a role in sodium-mediated fluid transport [89,90], and mutation in *SLC4A11* has been associated with a rare autosomal recessive disorder of the CE known as congenital hereditary endothelial dystrophy (CHED) [91]. Additionally, Type VIII collagen (*COL8A1* and *COL8A2*) is highly expressed in CECs and contributes to the composition of Descemet’s membrane, providing structural support for the corneal endothelium [92].

While the pluripotency and NCCs markers are well defined, identifying specific markers for CECs has been challenging. The commonly used markers for CECs, *ZO-1*, and *Na^+^K^+^-ATPase* (Figure 6A,B) are ubiquitously expressed; the hexagonal staining pattern, while typical, is not unique, and a similar pattern can be seen in many monolayer cells, such as lens and intestinal epithelium, and also in retinal pigmented epithelial cells [93,94,95,96]. Hence, there is a need for corneal endothelial-specific markers that can accurately represent corneal endothelial characteristics and functions [39], and especially so if the population of putative CECs are derived from a pluripotent cell source. In a study by Ding et al., two novel cell-surface monoclonal antibodies, TAG-1A3 and TAG-2A12, were generated using isolated cadaveric CECs. The antigen targets of TAG-2A12 and TAG-1A3 were identified as *sPRDX-6* and *CD166* (Figure 6C,D), respectively [22,97], which had not been previously described as markers for CECs. Interestingly, *sPRDX-6* was found to specifically bind the cell surface of CECs and not that of other cell types screened, including human pluripotent cells [97]. The binding specificity of *sPRDX-6* makes it a promising candidate for screening or characterizing iPSC-derived CECs.

To circumvent the issue of non-exclusive markers for CECs, researchers have conducted comparative studies comparing the expression profiles of induced CECs with primary human CECs. Using a mass spectrometry-based proteome sequencing approach, Ali et al. reported that 90.82% of the 6345 proteins identified in the primary CECs overlapped with the proteome of iPSC-derived CECs [59]. Hatou et al. examined the RNA profile of their Corneal Endothelial Cell Substitutes derived from iPSCs (CECSi) and compared it to primary CECs using DNA microarray analysis [61]. They found that the CECSi cells had undergone significant differentiation from iPSCs and exhibited characteristics consistent with human CECs [61].

### 4.4. Evaluating iPSC-Derived CECs in Animal Models

To date, numerous reports have described the differentiation and characterization of iPSC-derived CECs. However, in vitro studies alone are inadequate for clinical translation. In order to progress towards clinical trials, it is crucial to show proof of function of the iPSC-derived CECs within an animal model. Knock-in or knock-out mouse models serve as excellent tools for evaluating the potential of iPSC-derived CECs to restore corneal endothelial function in gene-defective disease phenotypes such as CHED and FECD [98,99,100]. By deleting or mutating the gene of interest, researchers can mimic the disease condition observed in humans. With a genome that is 99% similar to humans, mice offer a cost-effective model due to their small size [101]. Additionally, the relatively short lifespan of transgenic mice allows for the observation of treatment effects from birth through to old age, enabling researchers to quickly and safely test new therapeutic approaches in vivo [101]. However, due to the small size of mouse eyes, it will be challenging to evaluate the functionality of cellular therapeutics.

On the other hand, the rabbit model serves as a valuable corneal endothelial dystrophy model due to its larger eyes, which makes it more suitable for the evaluation of cellular therapeutics through surgical procedures [102], including corneal endothelial transplant. The size of the rabbit’s globe and the dimension of its corneal endothelium is similar to that of humans, allowing for the utilization of surgical instruments typically used in conventional corneal transplant procedures [102]. Furthermore, previous studies have reported that rabbit’s ECD, central corneal thickness, and corneal diameter decreased with age, similarly to the changes observed in human [102,103,104]. Sun et al., in a study where they transplanted iPSC-derived CECs into a rabbit model with a 7 mm CE defect [63], observed a reduction in corneal edema and a thinner central cornea thickness compared to that of the control group [63]. Despite its advantages and frequent use by research groups, it should be noted that the rabbit model may only be suitable for short-term study, due to the regenerative capacity of the rabbit CE [33].

As the U.S. Food and Drug Administration (FDA) recommends assessing cellular and gene therapeutic products in multiple animal models [105], the rabbit model can serve as a testing platform for evaluating the initial functionality of iPSC-derived CECs obtained from induction protocols. For longer-term follow-up to assess the efficacy and potential side effects of iPSC-derived CECs, a non-human primate (NHP) model such as the cynomolgus or rhesus monkey could be used. Moreover, NHPs share a high degree of genetic similarity and possess highly conserved protein sequences with humans [106]. Unlike the rabbit model, the CE of NHPs does not regenerate [102], hence enabling long-term follow-up after corneal endothelial transplant without the risk of repopulation by the animal’s own CE. In such a monkey corneal edema model, Hatou et al. transplanted the CECSi they derived, and observed improved corneal transparency, although some of the monkeys experienced mild to severe immunological rejections [61]. These findings show the importance of establishing robust animal models to evaluate iPSC-derived CECs, ensuring that immunological rejection risks are significantly minimized [45] before considering clinical applications.

Detecting tumorigenicity is a crucial step in verifying the purity of the iPSC-derived CECs. Immunodeficient NOD/SCID mice are commonly used in teratoma formation assays, which enable the detection of trace amounts of tumorigenic cellular impurities in human cellular therapeutic products [107]. Hence, the inclusion of such an animal model as part of the assessment process is necessary to evaluate the safety of iPSC-derived CECs. Traditionally, the sites of engraftment for the teratoma formation assay include the testis, liver, kidney capsule, hind leg muscle, and the subcutaneous space of the NOD/SCID mouse [108], and an alternative approach to inject the cells to be evaluated into the anterior chamber (AC) of the eye has been proposed to be more advantageous [109]. Specifically, the transparency of the cornea allows for the non-invasive monitoring of the injected cells without the need to sacrifice the animal, and tumour formation occurs more rapidly in the AC compared to the subcutaneous method [109]. Ingaki et al. reported a median tumour formation time of 18.50 weeks subcutaneously, whereas it was reduced to only 4.0 weeks in the AC [109].

## 5. Limitations

There remain several limitations that hinder the full utilization of iPSCs for generating CECs for transplants. The existing protocols vary in components that direct its differentiation through different signalling pathways, leading to limited reproducibility. Therefore, it is crucial to establish a robust and clinically applicable protocol [45].

The most promising approach for clinical application involves generating CECs from autologous iPSCs, which significantly reduces the risk of transplant rejection. However, this process is time consuming, considering the need to first generate the autologous iPSCs and subsequently direct its differentiation into CECs, potentially creating a heightened risk of genomic instability due to extended culture in vitro [45,110,111,112]. Such genomic instability of the iPSC-derived CECs will be a challenge for disease modelling and even more so for clinical applications. Furthermore, autologous iPSCs from patients with genetic corneal endothelial diseases (FECD and CHED) retain the underlying disease [45], which requires genetic manipulation to alter the disease-causing genes prior to differentiating CECs, hence it is more ideal to use healthy and HLA-matched iPSC cells from an iPSC bank [45].

An alternative to autologous iPSCs is the creation of an HLA haplobank, which stores stocks of clinical-grade iPSCs for HLA-matched allogeneic transplantation [64]. HLA matching, specifically HLA-A, HLA-B, and HLA-DR, is known to reduces allograft rejection [113,114,115] and diminishes the use of immunosuppressive drugs [113,116,117]. Studies have shown that approximately 90% of the Japanese population and 93% of the UK population can be covered by 140 [64,118] and 150 HLA-homozygous donors, respectively [119]. Therefore, inventorying iPSC stocks from HLA-homozygous donors can serve as a sustainable source of iPSCs.

Another proposal is to create universal iPSC lines where HLA matching is no longer critical for allogeneic iPSC transplantation. This is achieved by deleting the *B2M* and *CIITA* gene, which suppresses the expression of HLA class I and HLA class II, respectively [116,117,118]. However, deleting HLA makes the cells vulnerable to attacks from natural killer (NK) cells. Forced expression of NK inhibitory ligands, HLA-E and CD47, has been shown to help iPSC-derived cell types avoid NK-mediated killing [116,119,120]. In addition to universal iPSCs, semi-universal iPSCs can be generated through the HLA-C retaining method, where twelve haplotypes of HLA-C retaining iPSCs can cover 95% of the global population [116].

To ensure a reliable supply of high-quality and HLA-matched iPSCs, validated early passages of these iPSCs should be stored in a biobank, as it can help to address issues arising from the long-term culturing of iPSCs [45,120,121]. This provides a means to recover good iPSCs in the event that mutations arise whilst the process is ongoing. A similar approach should be implemented for iPSC-derived CECs, incorporating stringent quality checks at pre-determined checkpoints of the differentiation process [45,122].

Furthermore, residual undifferentiated iPSCs present another limitation. Whilst pluripotency markers are analysed to ensure that the presence of undifferentiated iPSCs in iPSC-derived CECs is minimal, current characterization methods via real-time PCR and flow cytometry are not fool-proof. The main concern with any residual stem cells is the potential formation of teratomas when iPSC-derived CECs are used in cell therapy [45]. Hence, teratoma formation assay [123] should be included as a rigorous assurance control alongside a proof-of-function animal model to demonstrate that the differentiation of iPSCs into CECs is complete, and the therapeutic outcome can be expected without the risk of tumorigenesis.

## 6. Translational Challenges

Due to the lengthy in vitro process involved in producing iPSC-derived CECs, cost is likely a major hurdle. The chemical components used for cell line maintenance, along with various quality control reagents are generally expensive. However, the cost of iPSC-derived CECs could be greatly reduced through the development of a robust and standardized protocol. Such a protocol should ensure high efficacy to justify the cost effectiveness of generating CECs from iPSCs.

To establish a gold standard protocol for corneal endothelial dysfunction cell therapy, it is crucial to address the current challenges posed by different protocols using diverse chemical components, small molecules, and multiple signalling pathways. The use of serum, Matrigel, and other animal-derived components in the differentiation media presents a major obstacle for the clinical application of iPSC-derived CECs in human subjects [39]. Variability in animal-origin components introduces additional complexities that may affect the reproducibility of the differentiation processes. Although xeno-free components are commercially available, they are often proprietary, lacking disclosure of the exact composition [39], which could pose regulatory issues in human trials. Despite these challenges, progress is being made in clinical applications, as there is an ongoing clinical trial (International Clinical Trials Registry Platform: JPRN-jRCTa031210199) using iPSC-derived CECs to treat bullous keratopathy [124].

Another important consideration is the compliance with good manufacturing practices (GMP). Due to the potential risks of xeno-contamination and infectious pathogens, it is not ideal to use iPSC-derived CECs generated under non-GMP conditions for any form of clinical trial or application [28]. GMP compliance must follow strict regulatory guidelines as defined by the local regulatory body where the protocol is being developed [28,125]. Although regulatory requirements will be arduous, and most likely differ between regions, the overarching goal remains to ensure the safety and quality of the developed protocol [28,125].

Immuno-histocompatibility between recipients and iPSC donors presents another hurdle. Establishing an HLA haplobank of iPSC stocks could be a solution, but choosing the right donor cell type is also crucial. Some cell types, such as blood cells or dermal fibroblasts, may carry more mutations and chromosomal abnormalities due to high turnover or exposure to ultraviolet radiation [44,126,127]. This is especially true for cells from older donors [44,126,127].

Additionally, iPSCs are known to retain residual epigenetic memory from their somatic cell source, which can bias their differentiation potential into specific cell types [44,128,129,130,131,132]. However, this epigenetic memory diminishes as cells are passaged [44,133,134]. Hence, the establishment of a biobank holding different passages of iPSC and differentiated CECs is important, as it provides a way to correct any deviated processes without having to start from scratch.

Graft survival is a major challenge in clinical applications [45]. It has been shown in a mouse syngeneic model that while undifferentiated iPSCs have been rejected, differentiated cells from iPSCs may be less immunogenic and not trigger immune rejection [44,135,136,137,138]. However, if the differentiated cells contain even trace amounts of undifferentiated iPSCs, the risk of immune rejection increases. Encouragingly, a clinical trial using autologous iPSC-derived retinal epithelial cells in patients with age-related macular degeneration has been conducted [55]. The study showed graft survival without the need for immune suppression during the 1-year study period [44,55]. Remarkably, no serious complications or signs of local or systemic disease were detected as a result of the transplant [55]. The immune privilege of the eye and the high success rate (up to 90%) of corneal allografts without HLA matching and systemic immunosuppressive drugs [139,140], suggest that the transplant of iPSC-derived CECs may only require topical immunosuppressants [141].

Despite the complexities involved, clinical trials involving pluripotent stem cells, particularly in ophthalmology, are increasing [142,143]. In 2019, the team of Nishida et al. reported the first successful transplant of iPSC-derived limbal stem cells for a female patient [144]. Following this, the team has treated three more patients and reported improved or improving eyesight for all trial participants [144].

## 7. Conclusions and Future Perspectives

The future of iPSC-derived CECs may involve HLA-matched iPSC-derived CECs to reduce the risk of allogenic immune rejection. Similar to the concept of a haplobank for iPSCs, a centralized facility for expanding, inducing, and storing HLA-typed iPSC-derived CECs can provide clinicians access to matching their patients’ haplotypes and shorten transplant waiting times. However, more research is needed to establish a harmonized approach for the derivation of CECs from iPSCs that is compliant to GMP requirements and potentially FDA regulations. This will include the establishment of a robust panel of markers and assays for its characterization, as well as safety and efficacy testing within suitable animal models for long-term observation to identify any risks of immune rejection or disease recurrence, all of which must be thoroughly established and rigorously tested before the use of iPSC-derived CECs can be translated to a clinical setting. Finally, a tri-party cooperation involving researchers, clinicians, and industry partners from the pharmaceutical or biotechnology sector with experience in cell therapy and large-scale process would be ideal [45]. Additionally, ophthalmic companies with expertise in developing and commercializing corneal therapies may also be valuable partners. This collaborative partnership will help bridge the gap between research and clinical application by providing resources, expertise, and regulatory knowledge with the aim of making iPSC-derived CECs a reproducible and affordable cell therapeutic product for patients with corneal endothelial dysfunction.

## Figures and Tables

**Figure 1 ijms-24-12433-f001:**
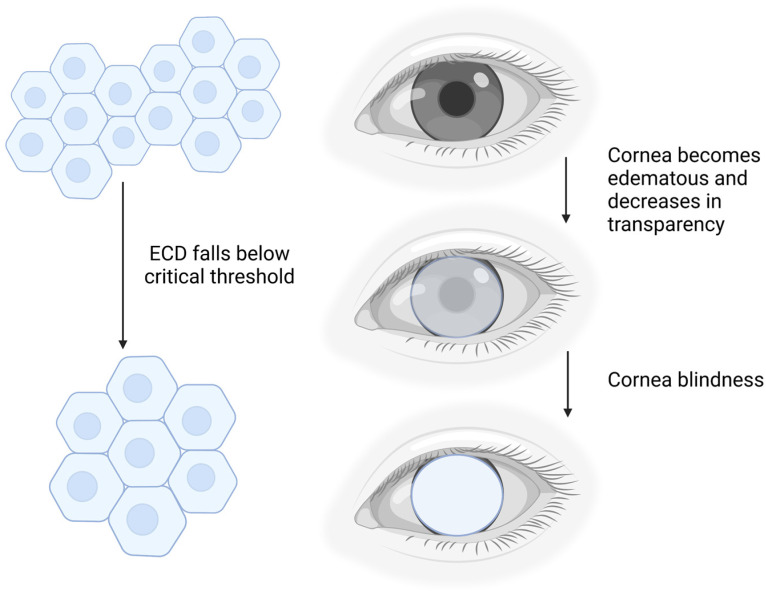
Schematic diagram of the effect of corneal endothelial cell density (ECD) on cornea opacity. Created with BioRender.com.

**Figure 2 ijms-24-12433-f002:**
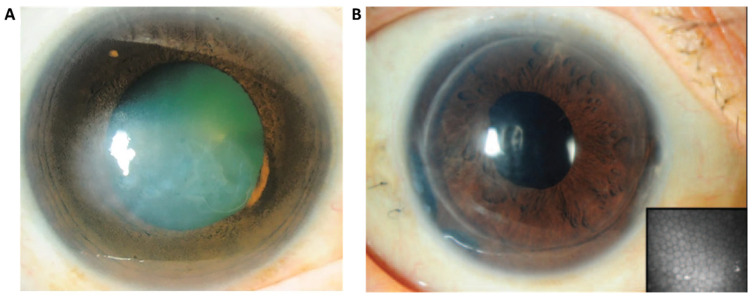
(**A**) Patient with corneal blindness due to severe corneal edema (bullous keratopathy) [12]. (**B**) 1-year post-DSAEK showing a clear cornea with healthy endothelial cells (20× magnification) [8].

**Figure 3 ijms-24-12433-f003:**
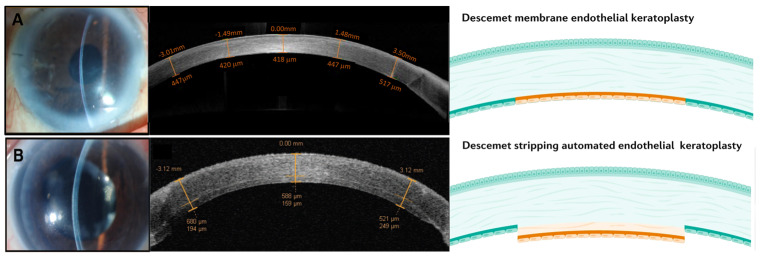
(**A**) Slit lamp image and anterior segment OCT (ASOCT) of DMEK patient [12]. (**B**) Slit lamp image and ASOCT of DSAEK patient [8,12].

**Figure 4 ijms-24-12433-f004:**
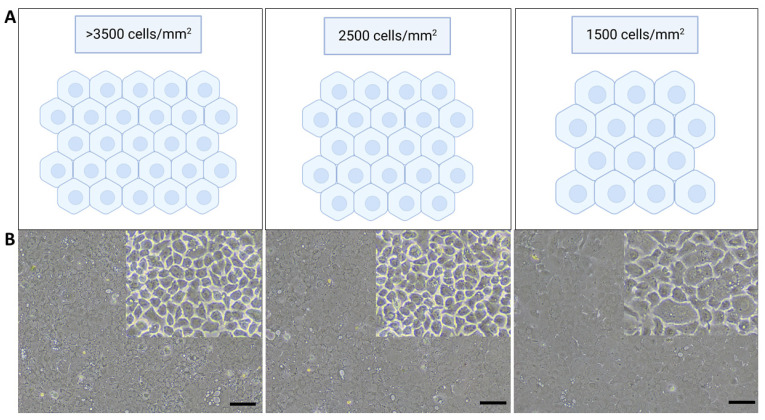
(**A**) Schematic diagram of corneal endothelial cells at different endothelial cell density (ECD). Created with BioRender.com. (**B**) Cultured primary human corneal endothelial cells (CECs) of different cell density at 10× magnification, taken using a Nikon Eclipse TS 100, Nikon Instruments, Melville, NY, USA [25].

**Figure 6 ijms-24-12433-f006:**
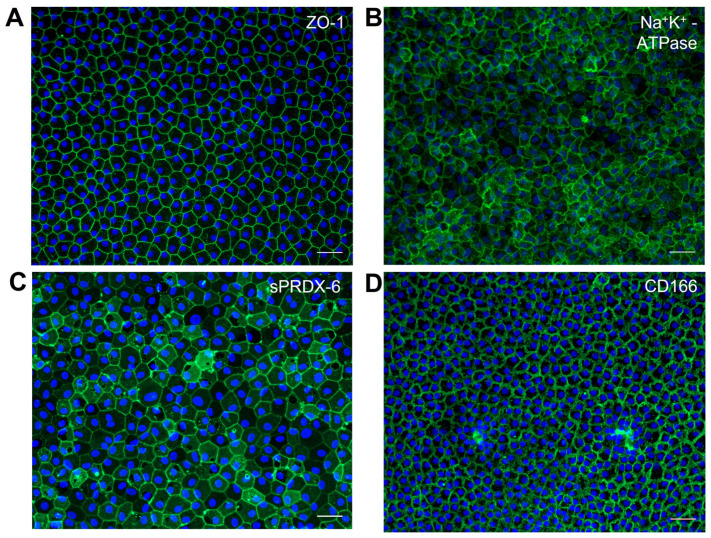
Cultured primary human corneal endothelial cells (CECs) at 20× magnification, taken on ZEISS Axio Imager M2m, Carl Zeiss Microscopy, LLC, White Plains, New York, USA. (**A**) Passage 2 CEC stained with Zonula occludens-1 (*ZO-1*) [28]. (**B**) Passage 2 CEC stained with *Na^+^K^+^-ATPase* [28]. (**C**) Passage 3 CEC stained with *sPRDX-6* [28]. (**D**) Passage 2 CEC stained with *CD166* [28].

## Data Availability

No new data were created or analyzed in this study. Data sharing is not applicable to this article.

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
