# Peer review of "Corneal Endothelial-like Cells Derived from Induced Pluripotent Stem Cells for Cell Therapy"

_ijms, 2023, doi:10.3390/ijms241512433_

Round 1

Reviewer 1 Report

Corneal Endothelial-like Cells Derived From iPSC For Cell Therapy

In the manuscript “Corneal endothelial-like cells derived from iPSC for cell therapy” the authors provide a comprehensive overview of the state of the art in corneal endothelial stem cell therapies.

It is a well written review and would be of considerable interest to those entering the field as well as a nice overview for those familiar with the field.   

Below you will find some point to point comments:

Background

Line 47: Suggest changing “contributed” to “contributes”

Line 64: Suggest changing “or at a lesser extent” to “or to a lesser extent”

Line 66: Suggest changing “in part form” to “in part from”

Line 67: Suggest changing “compounded with” to “compounded by”

Section 2

Line 94: Suggest changing “dysfunction CE” to “dysfunctional CE”

Figure 3 shows examples of a DMEK and DSAEK OCT but the DMEK image is a very early postop image. It shows some membrane detachment as well as a lot of edema. I would suggest changing it with a stabilised DMEK for comparison with a stable DSAEK.

Line 110: Suggest changing “associated to the age” to associated with the age”

Line 111: Suggest changing “grafts from donors fewer” to “grafts from donors with fewer”

Line 155: Suggest changing “the first via cell” to “the first is via cell”

Line 160: Suggest changing “and subsequently” to “and were subsequently”

Line 180: Suggest changing “injection offers the” to injection offer the”

Section 3

It seems a bit redundant to compare the pre2000 literature with the post 2000 literature regarding iPSCs given that the induction protocol was only published in 2006. I would suggest removing this comparison as it does not indicate expansion in the field rather the field pre- and post discovery. It could be replaced with a more granular view of the years since 2006 to reflect the actual increased interest in the field.

Section 4

This discussion of iPSC to NCC to CEC would be improved with cell image. Microscopy example images of the three next to each other would help the reader see the key morphological features of the cells on the pathway

While I understand that the development of CEC in utero is from NCC cells, is there evidence that this NCC step has considerable benefit when differentiating from iPSCs? Can you add any supportive data to inform the reader that the additional differentiation step confers considerable advantage over iPSC to CEC directly?

Again, in the case of 3D spheroids – I read in the text how it is done but why choose this method? Please add some information for the reader as to why 3D spheroids have advantages over the standard culture approaches.

Line 373: “found on corneal, lens and intestinal..” ZO1 is a tight junction protein found in a great number of epithelial and endothelial cells while Na/K pumps are found in all mammalian cells. This is what makes them poor as specific cell markers. It is their staining patterns that are typical (but also not unique or specific – see Van den Bogerd et al 2019 TVST – Corneal Endothelial Cells Over the past decade: Are we missing the Mark(er)?). It might be better to describe them that way rather than saying they are on lens and intestinal cells as examples.   

Section 5

Another point about using autologous iPSCs, is that in cases of genetic corneal endothelial disesese (CHED and Fuchs), using patients own cells still retains the underlying disease. This would require genetic manipulation of the iPSC to “cure the disease” prior to differentiating to CEC. Seems much more realistic, safer (and less expensive) to use an iPSC bank of healthy matched cells.

Section 6

No remarks

Section 7

No remarks

Author Response

We thank the reviewer for the suggestions and feedbacks, please see the attachment for our responses. 

Reviewer 2 Report

A stem cell-based regenerative medicine approach using induced pluripotent stem cells (iPSCs) offer a promising solution, as they possess self-renewal capabilities, can be derived from adult somatic cells, and can be differentiated into all cell types including corneal endothelial cells (CECs). In all, this review provides insights into the advancement and obstacles of using iPSC in the treatment of corneal endothelial dysfunction.

Author Response

We thank the reviewer for a favourable review.

Reviewer 2: A stem cell-based regenerative medicine approach using induced pluripotent stem cells (iPSCs) offer a promising solution, as they possess self-renewal capabilities, can be derived from adult somatic cells, and can be differentiated into all cell types including corneal endothelial cells (CECs). In all, this review provides insights into the advancement and obstacles of using iPSC in the treatment of corneal endothelial dysfunction.